

# Before the fire: Assessing post-wildfire flooding and debris-flow hazards for pre-disaster mitigation

Ann M. Youberg[1], Joseph B. Loverich[2], Michael J. Kellogg[3], Jonathan E. Fuller[3]

1Arizona Geological Survey, The University of Arizona, Tucson, 85721, USA
[2]JE Fuller Hydrology and Geomorphology, Inc, Flagstaff, 86001, USA
[3]JE Fuller Hydrology and Geomorphology, Inc, Tempe, 85284, USA

*Correspondence to*: Ann M. Youberg (ayouberg@email.arizona.edu)

**Abstract.** Increasing size and severity of wildfires, and the expanding built environment into the wildland-urban interface makes it imperative that local governments identify, prepare for and reduce risks to people and infrastructure from wildfires
and the aftermaths of fires. Here we report on a pre-wildfire assessment of post-fire hazards in Coconino County, Arizona, the mitigation measures identified and implemented as a result of the study, and proposed changes to the assessment methodology for upcoming studies. Results from the Coconino County study indicate that up to 34% of the buildings, and up to 26% of the critical facilities countywide are at some level of increased risk of post-fire flooding if no actions are taken to reduce the risk of severe wildfires. As many as 593 homes (2,191 parcels) in Coconino County, as well as 13 dams and other critical facilities,
may be impacted by post-fire debris flows. In two smaller areas of detailed study, flood peaks could increase as much as 4-5 times the existing 100-year flood levels, with up to a 350% increase in the number of buildings in flood-prone areas. Debris flows will likely be limited in aerial extent but could impact a much larger area from following floods and sediment-laden flows. Mitigation measures identified and implemented as a result of this study include Coconino County coordination with the Kaibab National Forest in regard to forest health projects, development of a post-wildfire emergency action plan for the
City of Williams, development of a Post-Fire, Pre-Disaster plan for the City of Williams and educating the City of Williams and County officials and business stakeholders of the post-fire flood risk. Other mitigation measures that are still in the planning stage include installation of additional early flood warning system gages, and increasing building and infrastructure resiliency through channel conveyance improvement and utility protection projects. For two upcoming assessments, we plan to use a new statistical methodology to develop burn severity maps using historical burns, and we plan to employ a new
process-based debris-flow model developed for use in Arizona to assess debris-flow inundation limits.

## 1   Introduction

In June 2010, the Schultz Fire burned 6,100 ha (15,075 ac) on the steep eastern slopes of the San Francisco Peaks northeast of Flagstaff, Arizona. This human-caused, wind-driven fire burned the majority of the area within the first 24 hours (USDA Forest Service, 2010). Rainfall from a wetter-than-normal monsoon caused post-fire debris flows on U.S. Forest Service lands and





extensive and repeated flooding in developed areas downstream, significantly damaging homes and infrastructure (Youberg et al., 2010). A full-cost accounting of the economic impacts from the Schultz Fire to the greater Flagstaff area is estimated between $133M and $147M (Combrink et al., 2013). Consequently, local governments and communities are now pro-actively undertaking projects to reduce risks of future wildfires and to identify and mitigate risks from post-fire hazards. For example,

in 2012, following two years of continued impacts to homes and infrastructure from the Schultz Fire, voters in the City of Flagstaff passed, with 75% approval, a $10M bond for city taxpayers to fund the Flagstaff Watershed Protection Program (FWPP). The FWPP pays for forest treatments on U.S. Forest Service lands to reduce risks of wildfires in watersheds that could threaten city water supplies or the city itself (Runyon, 2015).

In 2015, Coconino County launched a pilot project through the Federal Emergency Management Agency (FEMA) Cooperative

Technical Partners program to assess potential post-wildfire flooding and debris-flow hazards in the aftermath of a reasonable-scenario wildfire. Here, we summarize the results from this project; the complete report with appendices detailing methodologies and results can be found at the Arizona Geological Survey document repository (https://azgs.arizona.edu/; Loverich et al., 2017b). The purpose of the study was to help the County broadly identify areas at risk to post-wildfire hazards and to define, at the planning-level, the extent and severity of those risks in two selected study areas. The goals of the study

were to 1) conduct a countywide reconnaissance-level assessment to identify areas at greater risk of impacts from post-wildfire flooding and debris flows, 2) quantify those risks in two selected study areas and identify risk zones, and 3) identify mitigation opportunities to help reduce the risks before a fire occurs. Here, we also include a discussion of the challenges encountered during the project, plans for modifying our work procedures for an upcoming pre-fire assessment of post-fire hazards for Yavapai County, and some unexpected challenges encountered during planning and implementing of the identified mitigation

measures.

## 2   Study Area

Coconino County is located in northern Arizona; its county seat is Flagstaff (Fig. 1). Most of Coconino County is located on the Colorado Plateau physiographic province, with the southern edge extending below the Mogollon Rim into the Central Highlands Transition Zone. Vegetation communities vary with elevation and generally range from desert shrub in the lower

semi-arid deserts and canyons, grasslands and pinon-juniper woodlands on mid-elevation slopes, ponderosa pine and mixed conifer forests on the Mogollon Rim and lower to mid-mountain slopes, and spruce-fir forest and alpine tundra on the highest mountain slopes (Hendricks, 1985). Along Mogollon Rim, where the State's largest fires have burned (2002 Rodeo-Chediski and 2011 Wallow Fires), ponderosa pine forests are the dominant vegetation type. Geologic units in Coconino County are dominated by Palaeozoic and Mesozoic sedimentary rocks and middle to late Miocene to Holocene volcanic rocks of the San

Francisco Volcanic Field (Richard et al., 2000). The two study sites selected for detailed modeling and analyses were Fort Valley and the City of Williams (Fig. 1).



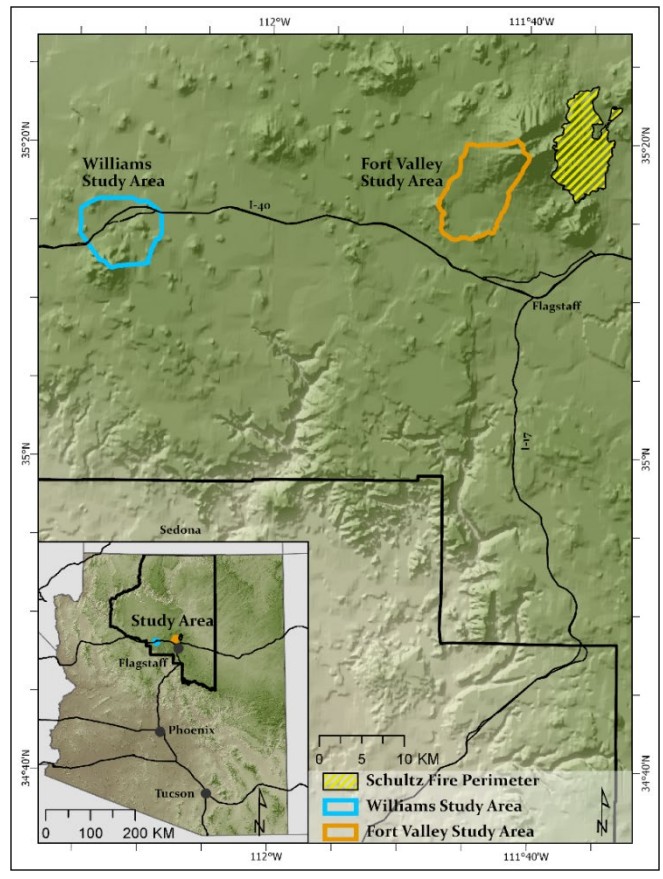

**Figure 1. Location map of Coconino County, Arizona (inset), the two study sites, Williams and Fort Valley, and the Schultz Fire burn area.**

Fort Valley is located in a low-gradient meadow setting at the southwestern base of the San Francisco Peaks (Fig 1). The San

5   Francisco Peaks are a multi-peaked Pleistocene stratovolcano (Wolfe et al., 1987; Holm, 1988). The upper part of the Peaks, above ~ 2460 m (8200 ft), is designated the San Francisco Wilderness area. The peak directly above and draining into Fort Valley is Agassiz Peak. Alluvial fans and active channels grade into ill-defined active flow channels and wide floodplains in a meadow setting. Fort Valley is not as densely developed as areas on the east side of the mountain that were impacted by the Schultz Fire but this is a popular and growing area.

10   The City of Williams (Williams) is located about 64 km (40 miles) west of Flagstaff (Fig. 1). The city was developed on flat-lying to low-gradient, late Miocene to Pliocene basalt flows (Richard et al., 2000) at the northern base of Bill Williams Mountain (BWM), which was formed by a cluster of Pliocene dacite, andesite, and benmoreite domes (Newhall et al., 1987). Cataract Creek has two primary tributaries originating on the south and east sides of BWM. Both tributaries have watersheds with heavily forested slopes and discharge into two municipal drinking-water reservoirs (City Reservoir and Santa Fe

15   Reservoir) before combining and flowing through the City.



## 3 Methods

We employed a multistep approach for this pre-fire assessment of potential post-fire hazards. First, using geographic information system (GIS) analyses, a countywide assessment was conducted to broadly identify areas that were (a) susceptible to wildfires, (b) populated or had significant infrastructure (e.g. railroads, dams, critical facilities, private residences, etc.), and

(c) were in a position topographically that made the area susceptible to post-fire debris flows and floods (Loverich et al., 2017b). Eight areas of concern were identified in this first step; two study areas then were selected for more detailed analyses based on the assessed risks and populations (Loverich and Kellogg, 2016).

Once the two study areas were selected, LiDAR data were collected to generate detailed topography for use in modeling. Risks in the study areas were assessed through field observations and modeling. Field observations within the study areas included

evaluating drainages for evidence of past debris flows (Youberg, 2016) and inventorying features in the developed areas to inform flood-flow modeling (Loverich and Kellogg, 2017). Wildfire modeling was conducted for the area above Fort Valley in order to create soil burn severity maps; these data and results from existing U.S. Forest Service (USFS) modeling on BWM were utilized to select appropriate runoff curve numbers for use in flood modeling (Loverich, 2016). Flood flows and debris flows were analysed using design storms with current and treated forest conditions to quantify impacts and (Loverich and

Kellogg, 2017; Youberg, 2017) and develop hazard zone maps (Loverich et al., 2017a). Finally, results and hazard maps were used to help identify potential mitigation measures (Loverich et al., 2017b).

### 3.1 Burn severity analysis

The purpose of modeling wildfire behavior under different burn scenarios (treated/untreated) was to (1) develop soil burn severity classification maps for use in flood and debris-flow modeling (Loverich and Kellogg, 2017; Youberg, 2017) and (2)

to select appropriate runoff curve numbers for use in flood modeling (Loverich, 2016). Prior to the start of this project, the Kaibab National Forest (KNF) had completed fire modeling of BWM to determine the crown fire potential as part of an environmental impact statement for the Bill Williams Mountain Restoration Project (USDA Forest Service, 2015a). Data from this modeling were used to generate soil burn severity maps for potential wildfires burning BWM under current (Untreated) and treated (Treated) forest conditions (USDA Forest Service, 2015a; Loverich, 2016).

The forested areas above Fort Valley are within the Coconino National Forest (CNF) and were not included in the modeled area for the adjacent Four Forest Restoration Initiative (4FRI). Thus, we followed similar procedures as KNF to model the crown fire potential using FlamMap Version 5. This model utilizes user-defined parameters to approximate fire behavior across a landscape under various scenarios (e.g. fuel moisture, forest characteristics, weather, etc.). We worked with experts from KNF, CNF, and 4FRI to select and modify model parameters to mimic conditions similar to the Schultz Fire (Lata, 2015;

USDA Forest Service, 2015a), and we assessed methodologies and interpretations from similar studies to guide our methodologies (Tillery et al., 2014; Runyon, 2015).



The Fort Valley study area lies at the base of the Agassiz Peak. The upper area of Agassiz Peak, above ~ 2460 m, lies within the San Francisco Peaks Wilderness area. Limits of activities within wilderness areas might make forest treatments problematic, thus, we modeled for three forest conditions: (1) Untreated - current forest conditions, (2) TreatedAll - forest treatments across all elevations, and (3) Treated8200 - treatments only outside the wilderness, below ~ 2460 m (~ 8200 ft)
(Loverich, 2016).

FlamMap provides various outputs that can be interpreted to develop burn severity maps, including crown fire potential and heat/unit area. Based on other local studies (Lata, 2015; Runyon, 2015; USDA Forest Service, 2015a), we used the Scott/Reinhardt crown fire calculation method to model Crown Fire Activity (active crown fire, passive crown fire, surface fire, no fire) which was used as a proxy for defining the soil burn severity levels (high, medium, low, unburned) (Loverich,
2016; Loverich et al., 2017b).

## 3.2  Flood analysis

We used a two-dimensional computer model to assess pre- and post-fire flows at each study site in order to understand and quantify the impacts of wildfires under current/untreated and treated forest (e.g. mechanical thinning, controlled burns, etc.) conditions on downstream flood flows. We used FLO-2D PRO, which allows for modeling distributary and unconfined
sheetflow over complex topography, and the 2-, 10-, and 100-year rainfall events to model flood flows at each site for three scenarios: (1) pre-fire existing conditions, (2) post-fire existing conditions (no treatments), and (3) post-fire treated conditions (Loverich et al., 2017b). Three scenarios were modeled in the Fort Valley study site to reflect the two potential treatments (TreatedAll and Treated8200) (Loverich and Kellogg, 2017).

This modeling effort also required the selection of appropriate runoff curve numbers to represent post-wildfire runoff. The
USFS often deploys a burned area emergency response (BAER) teams to help assess immediate impacts from a wildfire and to identify mitigation measures to reduce the impacts. In USFS Region 3, Arizona and New Mexico, BAER team hydrologists often use Wildcat5 software (Hawkins and Barreto-Munoz, 2016) to assess post-fire flood magnitudes. We used guidance from the Wildcat5 manual (Hawkins and Barreto-Munoz, 2016, Table 4-06) and results from our wildfire modeling to inform the selection of post-fire runoff curve numbers.

## 3.3  Debris-flow analysis

We used multiple models to assess the likelihood and impacts of potential debris flows in the study areas and to develop risk zones (Loverich et al., 2017b; Youberg, 2017). The probabilities of debris-flow occurrence for the study watersheds were assessed using the 2016 USGS logistic-regression model, M1 (Staley et al., 2016; Staley et al., 2017), and potential magnitudes were modeled using the 2014 USGS post-fire debris-flow volume model (Gartner et al., 2014). Both of these models use
parameters that describe basin topographic characteristics, which were derived in Esri ArcMap 10.4 using the 1 m LiDAR




digital elevation models (DEMs) and soil burn severity (Youberg, 2017). The wildfire modeling provided some, but not all, of the burn severity data needed for these models.

The soil burn severity data for the volume model were derived directly from the wildfire modeling which mapped crown fire activity into high, medium, low and unburned classification (USDA Forest Service, 2015a; Loverich, 2016). The probability

model, however, requires a continuous metric of soil burn severity that is typically provided after a wildfire by the differenced normalized burn severity ratio (dNBR) derived by comparing pre- and post-wildfire remotely sensed imaging (Key and Benson, 2006). Proxy dNBR values for the study sites were derived using the Esri ArcMap 10.4 zonal statistics tool with the continuous dNBR and classified soil burn severity data from the Schultz Fire (Youberg, 2017).

With the parameters for the probability and volumes models derived, we assessed the potential likelihood and magnitude of

post-fire debris flows in the study areas for the various current and treated forest conditions. Next, we re-arranged the probability equation, M1, to assess the peak 15-minute rainfall intensity ($I_{15}$) required to produce a debris flow with a 50% probability of occurrence to provide another comparison for quantifying the impacts of wildfires under current/untreated and treated forest conditions (Staley et al., 2017; Youberg, 2017).

To assess potential runout distances for estimating risk zones, we used mapped debris-flow deposits from the Schultz Fire

(Cook et al., 2017; Youberg, 2017) to gauge the efficacy of the USGS model Laharz_py (Schilling, 2014) for predicting inundation zones in this environment. We used the Schultz Fire deposits to test two sets of coefficients that describe flow mobility, one developed for worldwide debris flows (Griswold and Iverson, 2008) and the other for saturation-induced debris flows in southeastern Arizona (Magirl et al., 2010). Based on this assessment, we found the Arizona coefficients provided slightly better results in our study areas (Youberg, 2017). Laharz requires the user to select the point at which debris flows will

begin to deposit so we assessed characteristics of debris-flow basins in the Schultz Fire burn area to determine channel gradients where deposition occurred. Most debris-flow deposits were on channels with gradients between 5° and 10°, and mixed debris-flow and flood deposits were on channels with gradients from ≥ 2.5° to 5° (Youberg, 2017). We used this information, and channel confinement, to guide our selection of deposition points.

## 4   Results

Results from this study are summarized below. Detailed results are in a report by Loverich et al. (2017b; https://azgs.arizona.edu/).

### 4.1  Soil burn severity analysis

Burn severity modeling on BWM was conducted by the Kaibab National Forest for the Bill Williams Mountain Restoration Project as part of the environmental impact statement (USDA Forest Service, 2015a). Those results were used for the Williams

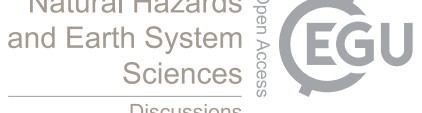



study area to create soil burn severity maps that represented a reasonable-scenario wildfire in untreated (Fig. 2a) and treated (Fig. 2b) forest conditions (Loverich, 2016).

Soil burn severity modeling for the Fort Valley study area was conducted using FlamMap Version 5 and the Scott/Reinhardt crown fire calculation method to model Crown Fire Activity. We modified the forest characteristics file to alter the mixed

5   conifer fuel load from light to heavy load to better mimic conditions during the Schultz Fire, and we altered the fuel load parameter to represent different forest conditions (Untreated/TreatedAll/Treated8200) (Loverich, 2016). These results were used to create burn severity maps for each of the modeling scenarios (Fig. 3).

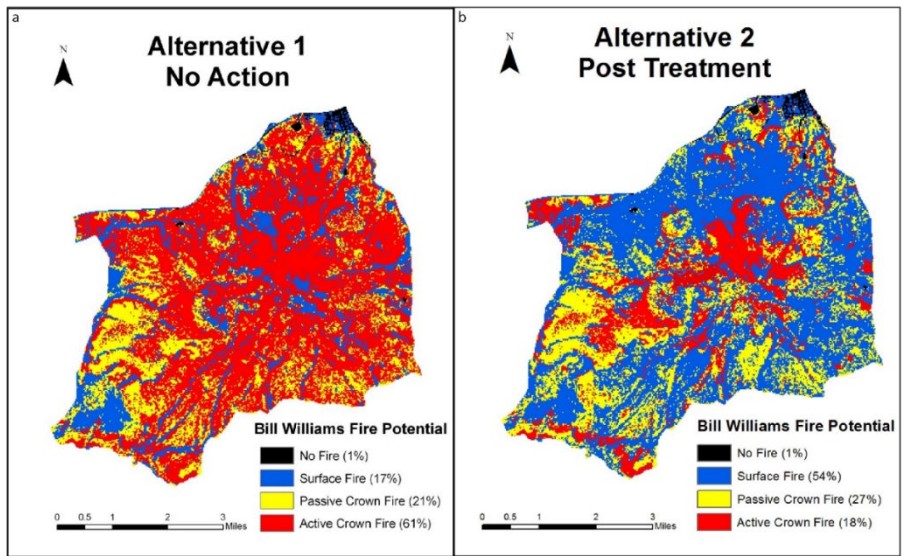

**Figure 2. Wildfire modeling and crown fire potential for BWM from the Bill Williams Mountain Restoration Project environmental**
10   **impact statement (from USDA Forest Service, 2015a). For the purposes of this project, Active Crown Fire, Passive Crown Fire, and Surface Fire represent high, medium and low soil burn severity, respectively, for Untreated (a) and Treated (b) forest conditions.**

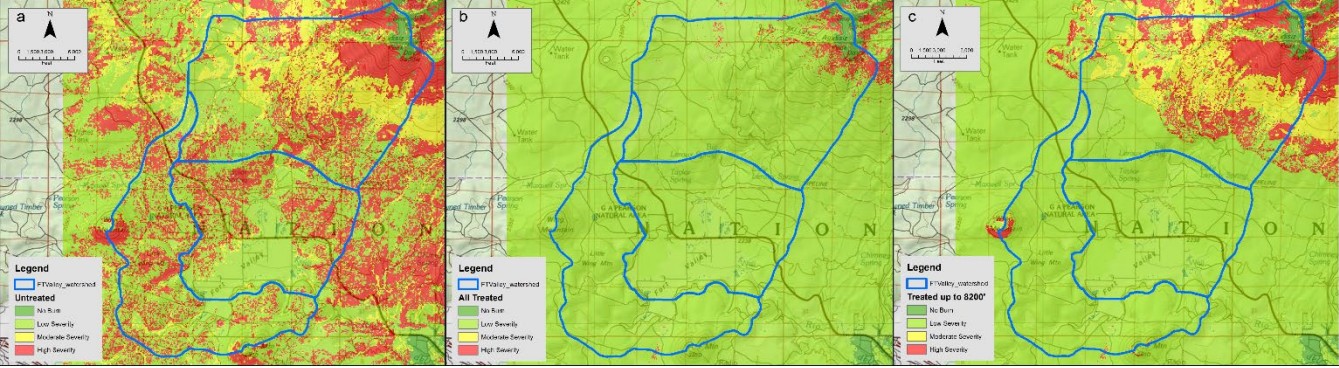

**Figure 3. Wildfire modeling and soil burn severity maps for the Fort Valley study area for Untreated (a), TreatedAll (b), and Treated8200 (c) (from Loverich, 2016)**



Results from the wildfire modeling were then used to select appropriate runoff curve numbers for flood flow modeling. We assumed that all areas burned at moderate and high severity had some water repellency and assigned curve numbers of 90 and 95, respectively (Hawkins and Barreto-Munoz, 2016, Table 4-06; Loverich, 2016).

### 4.2 Flood analysis

We modeled the 2-, 10-, and 100-year recurrence-interval rainfall events at each site, for current pre-fire conditions and for post-fire untreated and treated conditions to understand both the expected changes to flows as a result of forest treatments, and also at Fort Valley, the potential impacts from limitations to treatments. For the 100-year design storm, post-fire flood flows downstream of Fort Valley in the untreated (current conditions) scenario are up to four times higher than pre-fire flows (Figs. 4 and 5) (Loverich and Kellogg, 2017; Loverich et al., 2017b). Our analysis indicates that post-fire flows can be reduced by
58% if all areas are treated but decreases by only 28% if treatments occur only in non-wilderness areas (Loverich and Kellogg, 2017). Flows crossing Highway 180, in the middle of the study area (Fig. 5), are derived mainly from channels emanating from Agassiz Peak. Here, post-fire flood flows from the untreated scenario are up to eight times higher than pre-fire flows; our analysis suggest that flows may be reduced by up to 77% if the entire area is treated but only by about 15% if the wilderness is not treated, resulting in flows up to six times higher than pre-fire flows (Loverich and Kellogg, 2017). To understand how
these expected changes in flood flows could impact the developed areas, we compared the number of buildings impacted by depths > 30.5 cm (1 ft) for each scenario (Table 1) (Loverich et al., 2017b). None of these buildings are designated as critical facilities by Coconino County.

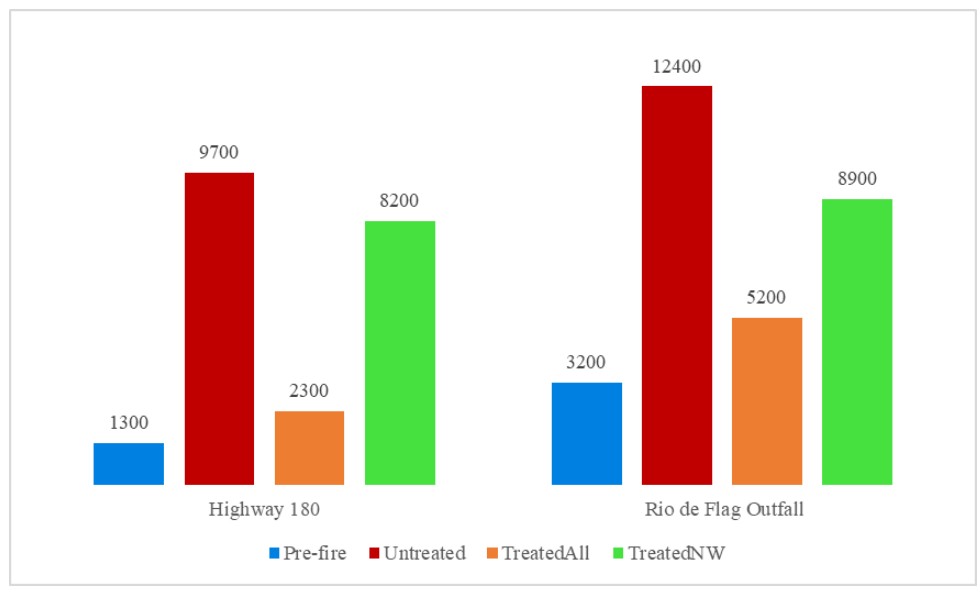

**Figure 4. Comparison of the 100-year pre- and post-fire flows (cfs) at Highway 180 and at the Rio de Flag for each forest condition**
**(from Loverich et al., 2017b).**





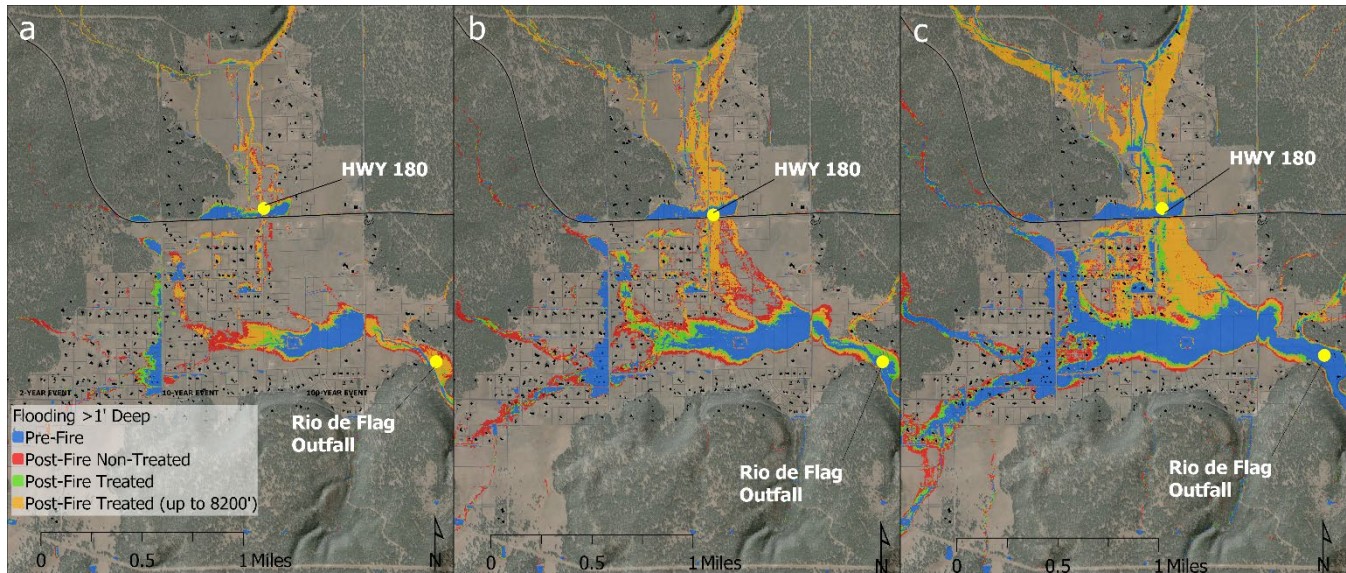

**Figure 5. Fort Valley flood-flow modeling results for each forest condition (Untreated, TreatedAll, Treated8200) comparing the pre-fire flows (blue) to post-fire flows from 2-year (a), 10-year (b), and 100-year (c) storms (from Loverich et al., 2017b)**

**Table 1. The number of buildings impacted in Fort Valley from each design storm and each scenario (from Loverich et al., 2017b).**

| Event | Pre-Fire | Post-fire No Treatment | Post-fire Treatment up to 8200' | Post-fire All Treated |
|-------|----------|------------------------|----------------------------------|-----------------------|
| 2-Year | 20 | 63 | 48 | 26 |
| 10-Year | 33 | 129 | 85 | 47 |
| 100-Year | 87 | 222 | 185 | 119 |

The City of Williams lies at the base of Bill Williams Mountain. Two watersheds of concern form the headwaters of Cataract Creek which flows northward through two municipal drinking-water reservoirs (City Reservoir, and Santa Fe Reservoir), and

10    then through the city itself. Model results from the 100-year design storm indicate increased post-fire flood flows at City Reservoir for the untreated (current conditions) scenario of up to eight times higher than pre-fire discharges, which based on our analysis may be reduced by 40% if the forest is treated (Figs. 6 and 7) (from Loverich et al., 2017b). At the south end of Williams, additional drainages join Cataract Creek and untreated post-fire flows are up to five times pre-fire discharges (Loverich and Kellogg, 2017). Our results suggest that forest treatments could reduce these flows by up to 49% which would





in turn reduce the number of buildings, including County-designated critical facilities, that would be impacted by flow depths > 30.5 cm (Table 2) (Loverich et al., 2017b).

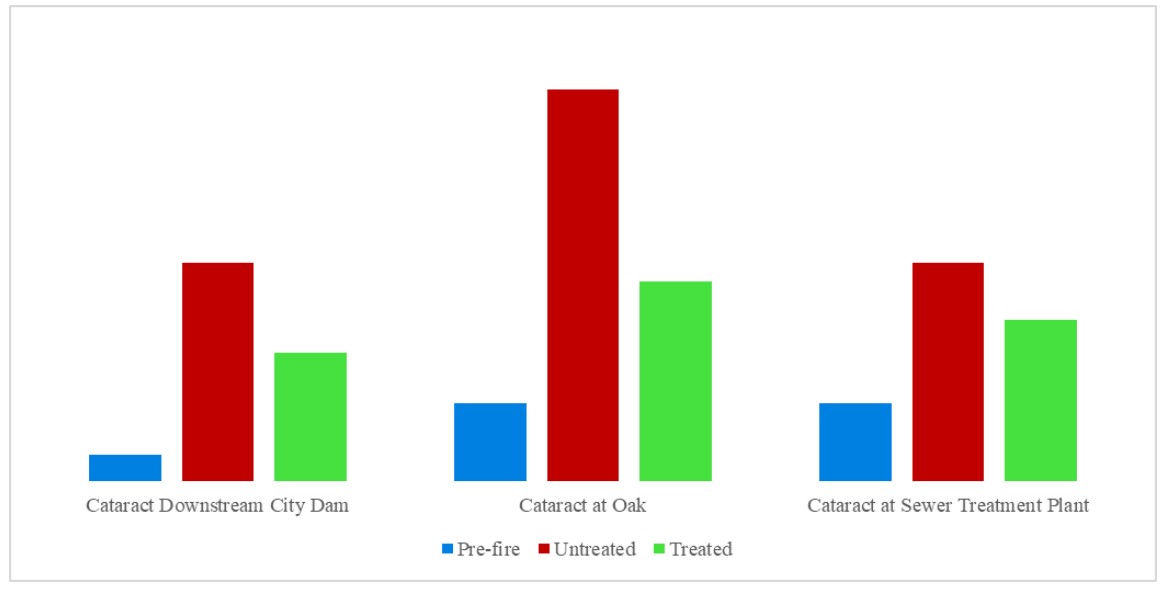

**Figure 6. Comparison of the 100-year pre- and post-fire flows (cfs) in the Williams study area for each forest condition (from Loverich et al., 2017b).**

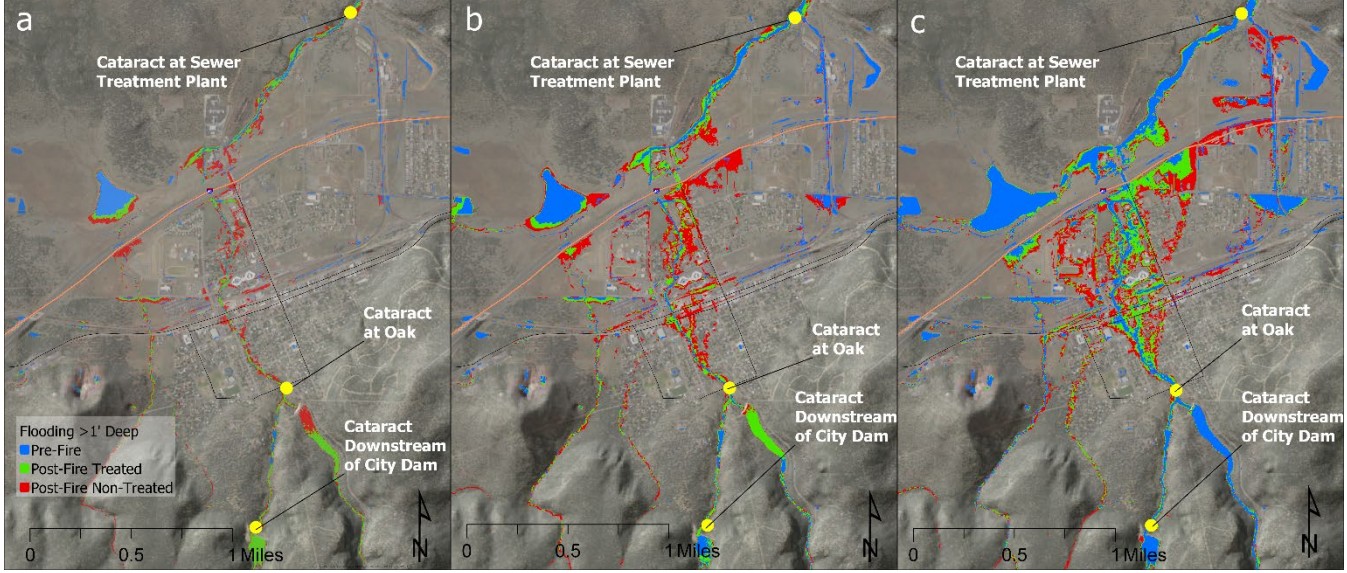

**Figure 7. Williams flood flow modeling results for untreated and treated forest conditions comparing the pre-fire flows (blue) to post-fire flows from 2-year (a), 10-year (b), and 100-year (c) storms (from Loverich et al., 2017b)**



**Table 2. The number of buildings (County-designated critical facilities) impacted in Williams from each design storm and each scenario (from Loverich et al., 2017b)**

| Event | Pre-Fire | Post-fire No Treatment | Post-fire Treated |
|---|---|---|---|
| 2-Year | 26 (1) | 117 (4) | 34 (1) |
| 10-Year | 41 (1) | 268 (7) | 105 (4) |
| 100-Year | 147 (4) | 515 (14) | 318 (8) |

### 4.3 Debris flow analysis

5   There is ample evidence of past debris flows from channels debouching into the Fort Valley meadow, and in drainages above the City of Williams, especially on Cataract Creek (Youberg, 2016). Ten basins on the southwest side of the San Francisco Peaks that drain into Fort Valley study site and 22 basins on BWM that drain into Williams were modeled for debris-flow probability (Staley et al., 2016), and volume (Gartner et al., 2014) using 1-, 2- and 5-year recurrence-interval design storms. Combined hazard rankings for each basin were based on probability and potential magnitude (Youberg, 2017).

10   Probability and volume results from a 1-year design storm for the Fort Valley basins are lower for the TreatedAll scenario but similar for the Treated8200 and Untreated scenarios (Fig. 8, Table 3) (Loverich et al., 2017b). The peak $I_{15}$ rainfall intensity needed to produce a 50% probability of debris-flow occurrence ranges from 20-81 mm h$^{-1}$ (0.8-3.2 in h$^{-1}$) with recurrence intervals that are generally < 1-year but range to 5-years for the TreatedAll scenario (Table 3) (Loverich et al., 2017b).

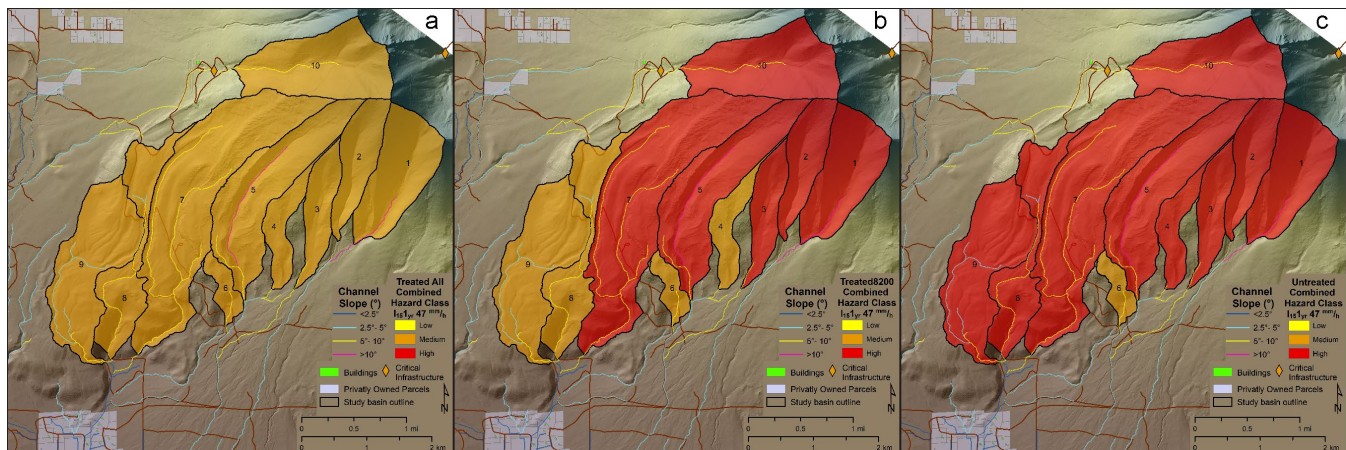

**Figure 8. Debris-flow combined hazard class (probability + volume) for a 1-year design storm for the Fort Valley basins for TreatedAll (a), Treated8200 (b), and Untreated (c) forest conditions (from Loverich et al., 2017b)**



**Table 3. Range of debris-flow probabilities and volumes from a 1-year design storm, and peak I$_{15}$ rainfall intensity for producing a 50% probability of a debris flow and the associated approximate recurrence interval (from Loverich et al., 2017b; Youberg, 2017).**

| Modeled Scenario | Probability of Debris Flow in 1-Year Event | Modeled Volumes (m$^3$) | Peak 15-minute Rainfall Intensity (cm h$^{-1}$) | Approximate Storm Recurrence Interval |
|---|---|---|---|---|
| TreatedAll | 45% - 77% | $10^3 - 10^4$ | 37-81 | 1 – 5-year storm event |
| Treated8200 | 66% - 99% | $10^3 - 10^{4.5}$ | 2046 | <1-year storm event |
| Untreated | 77% - 99% | $10^3 - 10^{4.5}$ | 20-37 | <1-year storm event |

Probability and volume results from the 1-year design storm for the Williams basins are also lower for the treated scenario
5  compared to the untreated condition (Fig. 9, Table 4). The peak I$_{15}$ rainfall intensity needed to produce a 50% probability of debris-flow occurrence ranges from 11-47 mm h$^{-1}$ (0.4-1.9 in h$^{-1}$) with recurrence intervals that are generally ≤ 1-year for both the untreated and treated scenarios (Youberg, 2017).

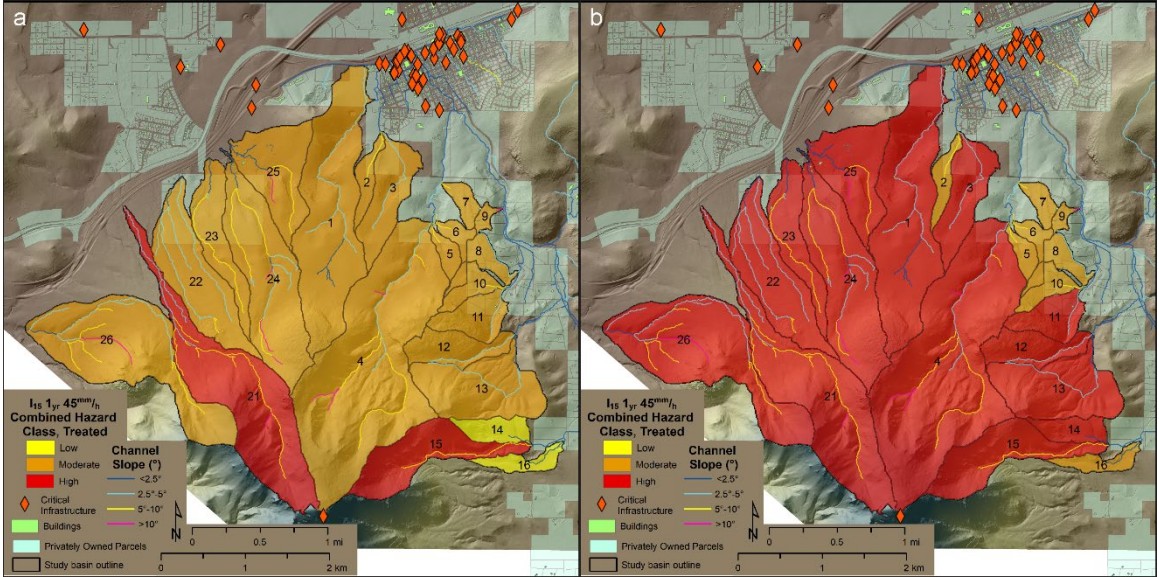

10  **Figure 9. Debris-flow combined hazard class (probability + volume) for a 1-year design storm for the Williams basins for Treated (a) and Untreated (b) forest conditions (from Loverich et al., 2017b)**



**Table 4. Range of debris-flow probabilities and volumes from a 1-year design storm, and peak I₁₅ rainfall intensity for producing a 50% probability of a debris flow and the associated approximate recurrence interval (from Loverich et al., 2017b).**

| Modeled Scenario | Probability of Debris Flow in 1-Year Event | Modeled Volumes ($m^3$) | Peak 15-minute Rainfall Intensity (inches/hour) | Approximate Storm Recurrence Interval |
|---|---|---|---|---|
| Treated | 38% - 94% | $10^3 - 10^4$ | 27-47 | ≤1-year storm event |
| Untreated | 66% - 99% | $10^3 - 10^{4.5}$ | 11-44 | <1-year storm event |

Volumes from the USGS model guided our choice of half-order magnitude volumes, from $10^3$ to $10^5$ $m^3$, for Laharz debris-flow inundation modeling (Youberg, 2017). Multiple deposition points were used on some channels to capture a wider range of areas where deposition could occur based on channel slope, confinement and potential avulsions. Modeled inundation zones with Laharz volumes of $10^3$, $10^{3.5}$ and $10^4$ $m^3$ most closely approximated USGS volume model results and appeared reasonable based on topography while the two larger volumes, $10^{4.5}$ and $10^5$ $m^3$, resulted in topographically unrealistic depositional patterns (Fig. 10) (Youberg, 2017).

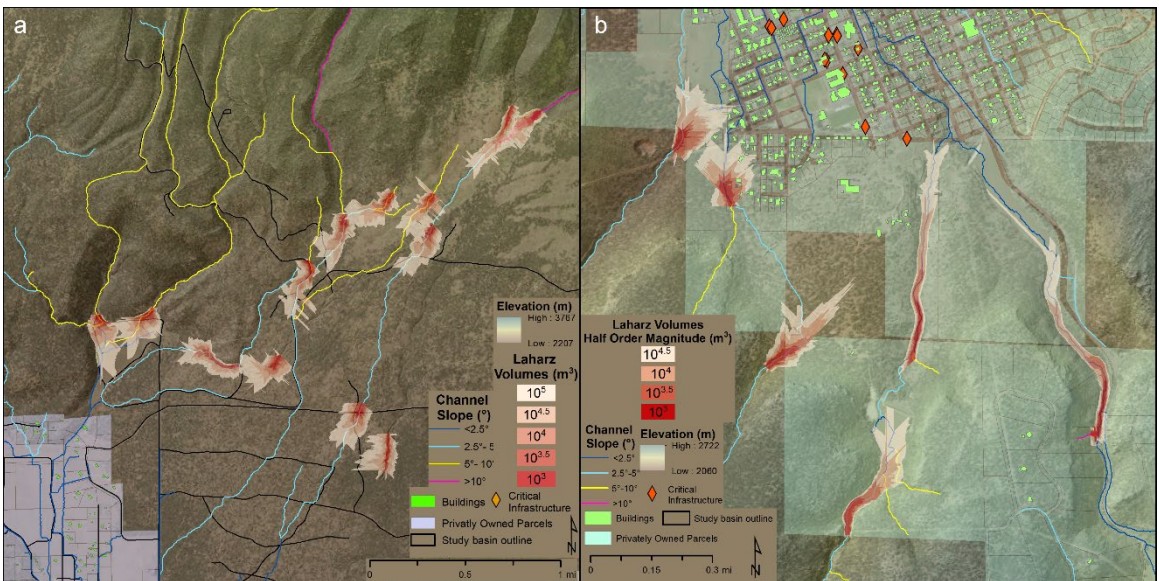

**Figure 10. Laharz model results for some of the Fort Valley (a) and Williams (b) basins for volumes of $10^3 – 10^5$ $m^3$ (from Youberg, 2017).**

### 4.4 Non-regulatory risk zone maps

The findings from the pre- and post-fire flood and debris-flow assessments are summarized in non-regulatory risk zone maps for the 2-, 10- and 100-year events (Fig. 11) (Loverich et al., 2017b). Four risk zones were defined by conservatively combining





model results for each hazard. Existing Conditions Flood represents current forest conditions and flood depths > 1 foot, although shallow flood limits may extend beyond the presented zone. Potential Post-Fire Flood represents areas that could be inundated by > 1-foot flow depths for untreated forest conditions. Post-Fire Debris-Flow represents a first-order approximation of an area were multiple debris-flow deposits from multiple events are most likely to occur. Post-Fire Hyperconcentrated Flow

5 represents an area between the debris-flow and flood zones where sediment-laden flows, erosion and sediment transportation and reworking are likely to occur (Loverich et al., 2017b).

In the Fort Valley study area, debris flows are less likely to directly impact the built environment but hyperconcentrated flows and flood flows could significantly impact homes and infrastructure, similar to the post-Schultz Fire flows. In Williams, debris flows could directly affect the built environment, but with a limited impact due to limited potential volumes and abrupt decrease

10 in channel gradient (Youberg, 2017). Potentially the greatest hazard from debris flows in Williams is in Cataract Creek where debris flows could fill City Reservoir (Youberg, 2017). The most significant risks to Williams, however, are from floods (Loverich et al., 2017b).

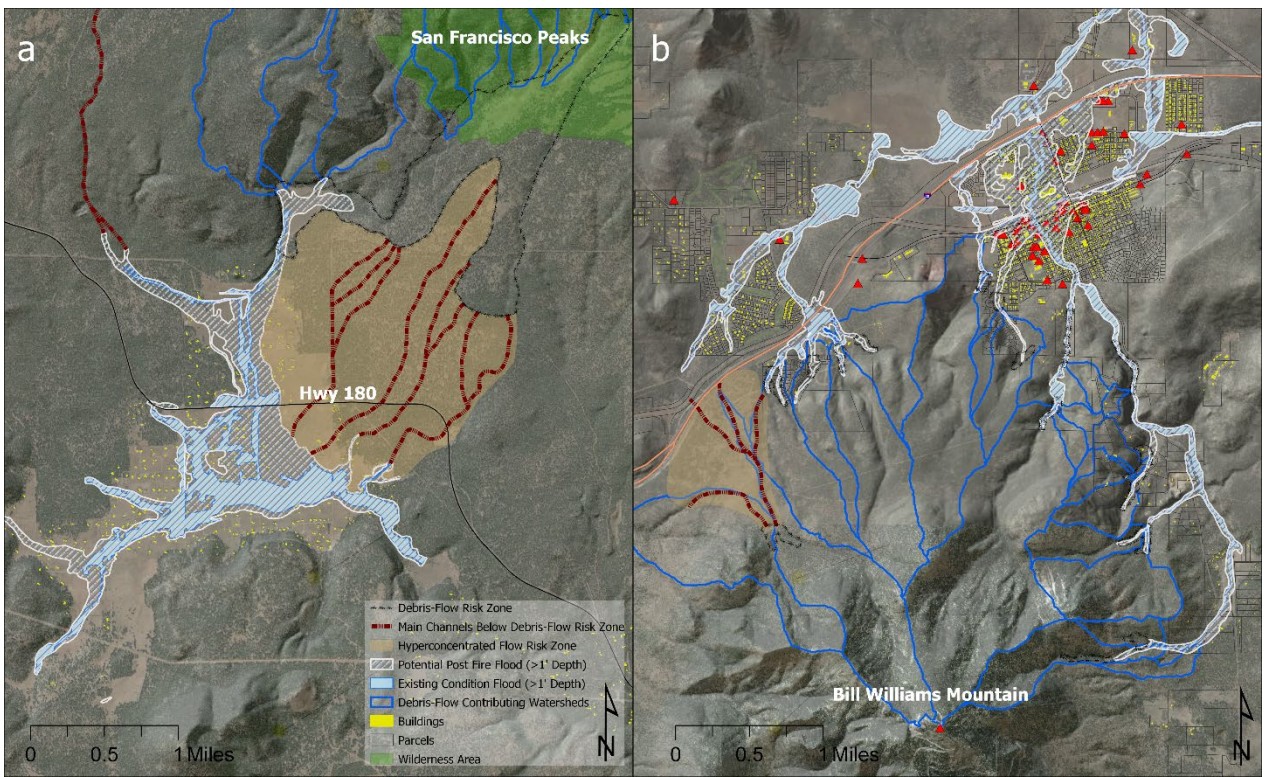

15 **Figure 11. Non-regulatory risk-zone maps showing the combined hazards for the 100-year event for Fort Valley (a) and Williams (b) (from Loverich et al., 2017b).**





## 5    Mitigation opportunities identified

The purpose of this study was not only to determine the potential post-wildfire hazards impacts, but to also to identify possible mitigation efforts that can be implemented now, before a fire occurs, to increase resiliency of the impacted communities. There are two primary ways to mitigate risks from post-fire hazards. The first is to implement forest treatments, such as thinning and

control burning, to either limit the probability of a fire, or to limit the severity of a fire by limiting fuels. The second is to identify and implement improvements or protection measures before a fire occurs and/or the forest is treated. These two mitigation methods should be considered together. Mitigation opportunities identified for both the City of Williams and Fort Valley study areas include forest health and restoration initiatives, development of a post-wildfire emergency action plans, community outreach and education, augmentation of the current flood warning system through installation of additional rainfall

gages within vulnerable watersheds, development of development guidelines for areas with a post-fire flood potential, and increasing building and infrastructure resiliency in post-fire floodprone areas (Loverich et al., 2017b). While mitigation measures were identified for both study areas, the potential impact of implementing those measures are limited in the Fort Valley area due to limitations of forest treatments in the wilderness area.

The City of Williams, however, has a very high flood risk if a wildfire were to occur on Bill Williams Mountain and there is

an opportunity to mitigate those risks. Prior to this study, there was an understanding within local governments that a significant post-fire flood risk existed in the City, but the possible extent and severity of the potential flood risk to the city identified in this study was unexpected and quite staggering. Not only is there a risk to life and property, but there will be a very large economic impact. The NAU Rural Policy Institute prepared an economic impact analysis using the results from this study and estimated post-fire flooding impacts to the City of Williams to be between $379,000,000 and $694,000,000 (Combrink and

Rousse, 2018). This has provided the motivation and opportunity for unusual collaborations among government agencies and with the non-profit sector.

Results from this study has helped initiate an ongoing coordination effort between the KNF, Coconino County, and the City of Williams to perform forest treatments on BWM, and increase the resiliency of the City of Williams should a fire occur. For the last 10 years, the KNF has worked towards getting environmental approval of forest treatments on the mountain as a part

of the Bill Williams Mountain Restoration Project. The final Record of Decision for the project was issued in December of 2015 (USDA Forest Service, 2015b), and the KNF has been taking the initial steps to thin portions of the project. The effort however has been limited by resource and funding availability as many of the acres of proposed treatment are on steep slopes that require very expensive methods, such as helicopter logging.

Based on the experiences from the 2010 Schultz Fire, results of this study and the economic impact study, the Coconino County

Flood Control District has identified post-fire flooding as the greatest single risk to the County (Coconino County Board of Supervisors meeting, 11 Jan 2019, item 23, http://coconinocountyaz.swagit.com/play/01112019-693). This identification has allowed Coconino County the option of utilizing Flood Control District money to fund treatment of sections of forest that, if



they were to burn, pose the greatest post-fire flood risk to downstream communities. Coordination between the County Board of Supervisors and Forest Service has allowed for unique collaboration and there are current proposals for the County to assist in funding some of the treatments through coordination with the National Forest Foundation, a non-profit contracting partner to the Forest Service.

Simultaneously, Coconino County and the City of Williams have been working, with assistance from outside resources, to develop a Post-Fire, Pre-Disaster Plan for the City. Post-fire flooding from Bill Williams Mountain has the potential to impact homes, businesses, critical infrastructure, transportation corridors, water supply sources, and utilities. Due to the high potential runoff rates and high sediment, ash, and debris concentration of post fire floods, impacts to the town could be in the form of flooding, erosion, and structural damage. Close coordination with the City of Williams, Coconino County, Kaibab National

Forest, and other major public agency and private business stakeholders within the area has enabled the technical team to identify ways to limit the potential damage of post-fire flooding. The plan identifies several ways in which the City can begin to increase resiliency and take steps to plan for a post-wildfire event through: structure protection, erosion protection, channel conveyance improvements, water source protection, reservoir management, and emergency action planning. The identification of these measures and specific infrastructure needs provides a significant step forward for the City of Williams and Coconino

County to receive FEMA funds through several programs (e.g. Pre-Disaster Mitigation, Flood Mitigation Assistance, Hazard Mitigation Grant Program) to implement necessary construction to reduce risks. Another critical component of preparing for a fire and the potential flooding impacts is pre-determining emergency actions and procedures. To that end, an Emergency Action Plan was also developed and is proposed to be tested as a Table Top Exercise.

## 6    Discussion

Communities that have experienced post-fire flooding and/or debris flows understand that a wildfire can signify the beginning of the damage and risk to life and property in downstream areas. Obvious as it may be, the cause of post-fire flooding and debris flows is a rain event over a recently burnt watershed. Not all burnt watersheds, however, produce a similar post-fire flood or debris-flow risk. As this study has shown, the severity of risk to life and property is variable based on community locations, upstream topography, surrounding vegetation, and current forest health. In the case of the 2010 Schultz Fire, post-

fire debris flows were limited to US Forest Service lands, but the post-fire flooding caused significant damage to downstream neighborhoods, well beyond the burn scar; as with most areas, the potential post-fire risks were unknown prior to the fire.

The question that many communities wrestle with is how to plan for and mitigate post-fire hazards before they happen. Ideally, communities and public agencies have the opportunity to identify areas with post-fire hazards, determine the extent and severity of those risks, and implement mitigation measures prior to a wildfire, not only reducing the probability of a fire

happening but also reducing the risk of post-fire hazards if a fire does occur. Mitigation of post-fire hazards before a fire, however, is not possible until the extent and severity of post-fire hazards are fully understood. This study is the first step in





understanding which areas in the Coconino County are most at risk for post-fire flooding and debris-flows, if a fire occurs in the contributing watershed. During the countywide, reconnaissance-level assessment, eight areas of concern were identified as having significant risk of post-fire flooding or debris flows, should a fire occur. Two locations stood out as posing the most risk to life and property and were selected for more detailed study, Fort Valley and Williams.

Both the countywide, reconnaissance-level assessment and the detailed assessment of the two study areas posed challenges. We had to develop a series of steps to identify, at a countywide reconnaissance-level, areas at risk from post-fire hazards. We used a GIS with topographic, vegetation and built-environment data to systematically assess and identify, in a repeatable manner, areas of concern (Loverich and Kellogg, 2016). We intend to use this methodology on two upcoming pre-fire assessments of post-fire hazards in Arizona. While several studies have assessed the likelihood of post-fire debris flows (Elliott

et al., 2011; Tillery et al., 2014; Tillery and Haas, 2016; Staley et al., 2018), to our knowledge there are no pre-fire assessments of post-fire hazards that also model the potential flood and debris-flows to identify downstream risk zones. We used several models, which presented their own challenges, to assess the hazards and combined the results to delineate downstream planning-level risk zones.

Fire modeling with FlamMap is a complicated process and is subject to user interpretation. As used here, and in other studies,

Crown Fire Activity is assumed to be a proxy for soil burn severity. These results provide a classified (high, medium, low) burn severity which was adequate for the flood-flow and debris-flow volume modeling, but the USGS post-fire debris-flow probability model requires a continuous dNBR value (Staley et al., 2016). Here we used Schultz Fire data to approximate dNBR values, but in upcoming studies we plan to use the methodology of Staley et al. (2018) to statistically develop soil burn severity for the study areas using remotely sensed data from nearby historical wildfires. This methodology will not only provide

statistically derived, continuous dNBR values and classified soil burn severity, it should also provide more reasonable, reliable and repeatable results.

The flood analyses were conducted using FLO-2D PRO, which allows for modeling distributary and unconfined sheetflow over complex topography. This model worked well because high-resolution topographic data existed for the low relief of our study areas. For our study, Coconino County was able to acquire airborne LiDAR data for the study sites which provided 1-m

resolution DEMs. Many communities, unfortunately, may not have the funds to acquire this level of data which will impact how well flood-flow risk zones can be defined.

The flood-flow modeling also showed the impacts of forest treatments. In the Fort Valley study area, flood flows were not significantly reduced when forest treatments were only applied to areas outside the wilderness as these areas constitute the lower slopes of the mountain. In the Williams study area, forest treatments applied to all elevations of the mountain will help

significantly reduce post-fire flood risks.

The debris-flow analyses were conducted using USGS models for post-fire debris-flow probability (Staley et al., 2016) and volume (Gartner et al., 2014) and Laharz_py (Schilling, 2014) for inundation modeling. The probability and volume models





were developed specifically for post-fire assessments, and data from the Schultz Fire were used to test the efficacy of the probability model (Staley et al., 2016; Staley et al., 2017). Results from these analyses show that in both study sites post-fire debris flows can be generated from very little rainfall. For the Fort Valley study area, the models show that when the entire forest is treated (TreatedAll), there is a decrease in both post-fire debris-flow probability and volumes. When treatments are

applied to areas only outside the wilderness (Treated8200), there is no significant difference between this and untreated conditions since the debris-flows originate on the steeper slopes, primarily within the wilderness area. In the Williams study area, treatments reduced debris-flow probability and volume but did not significantly increase the amount of $I_{15}$ rainfall necessary for a 50% probability of a debris flow. This is likely due to the wildfire modeling which resulted in large, contiguous patches of high soil burn severity on the steeper slopes, where debris flows are generated, for both treated and untreated

conditions.

Different forest conditions were not considered for the debris-flow inundation modeling needed to develop downstream risk zones, but results from the USGS volume model informed the selection of Laharz volumes. Laharz was developed to model volcanic and saturation-induced debris-flow inundation zones but it has not been commonly used in the post-fire environment where debris-flow area triggered by runoff (Kean et al., 2013; McGuire et al., 2017). Indeed, there are no Laharz mobility

coefficients for post-fire debris flows. Here, we tested Laharz using Schultz Fire debris-flow data (Cook et al., 2017; Youberg, 2017) and mobility coefficients developed from saturation-induced debris flows in Arizona (Magirl et al., 2010). For this study, we determined that results from Laharz provided adequate data to develop first-order approximations of post-fire debris-flow risk zones. Future studies, however, may benefit from new but ongoing efforts to develop a process-based inundation model for post-fire debris flows in Arizona, and which may also lead to the development of appropriate Laharz post-fire debris-flow

mobility coefficients (Youberg and McGuire, Accepted).

Results from the flood and debris-flow modeling efforts were combined to define non-regulatory risk-zone maps (Loverich et al., 2017a). The development of these maps also presented a challenge in determining how best to represent potential risks from events that have so many modeling components. Because we had high-resolution digital topography and experience with FLO-2D and the post-Schultz Fire floods, the flood flow zones are probably fairly well constrained. The debris-flow and

hyperconcentrated-flow risk zones were conservatively delineated based on model results and post-Schultz Fire flows, but should be considered first-order approximations. And finally, an unexpected challenge came during implementation of identified mitigation measures. We had not anticipated some of the difficulties local entities have encountered with some of these measures. In future studies, we will attempt to involve local agencies and stakeholders during the task of identifying mitigation measures with the hope of eliminating, or at least reducing, barriers to implementation.



## 7    Conclusions

Increasing size and severity of wildfires, and the expanding built environment into the wildland-urban interface makes it imperative that local governments identify and prepare for and reduce risks to people and infrastructure from wildfires and the aftermaths of fires. We summarized our findings from a pre-fire assessment of potential post-fire hazards project conducted for Coconino County, Arizona. The purpose of the project was to identify, at the countywide scale, areas at risk of post-wildfire flooding and debris flows, and to quantify the severity and extent of those risks within two study areas. We used other studies to help guide our methodologies for modeling burn severity, and for modeling floods and debris flows from burned areas, but we developed our own methodologies to define, at the planning-level, downstream risk zones within the study areas. In the Fort Valley study area, we showed that forest treatments applied to areas only outside of the wilderness have little impact on reducing post-fire flood and debris flows. In the Williams study area, we showed how significant the impacts of post-fire flooding could be to the City if a wildfire burns before the forest is treated. This has led to new collaborations between multiple government agencies and also the non-profit sector. We encountered several challenges during this project, and have identified ways to reduce those challenges in future projects. Coconino County and its residents have benefited from the County's proactive approach of identifying and mitigating areas at risk to post-wildfire hazards.

**Data Availability.** The full report with appendices is available for download from the Arizona Geological Survey Document Repository (https://azgs.arizona.edu/).

**Author contributions.** JL and MK lead the countywide assessment with assistance from AY and JF. JL and MK conducted the hydrologic modeling and AY conducted the debris-flow modeling. JL was lead author for the risk maps and the final report with contributions from all authors. AY and JL wrote this original draft with contributions from ML and JF.

**Competing interests.** The authors declare that they have no conflicts of interest.

**Acknowledgements.** This work was supported by Coconino County Flood Control District through a FEMA Cooperating Technical Partners (CTP) Program. The Arizona Geological Survey also provided assistance. The full project report with appendices can be found at the Arizona Geological Survey Document Repository: https://azgs.arizona.edu/. We also thank the Coconino and Kaibab National Forests for their assistance on this project, especially Mary Lata, Christopher MacDonald, Tom Runyon and Mike Uebel.



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
