# Peer review of "Before the fire: Assessing post-wildfire flooding and debris-flow hazards for pre-disaster mitigation"

_Natural Hazards and Earth System Sciences, 2019_

## Referee Comment (RC1) · Anonymous Referee #1 · 23 Apr 2019

This manuscript does not conform to the stated format of the journal, which asks for "...original research on natural hazards and their consequences." While the writing and figure production is of high quality, this is not an original piece of research and unfortunately is not fit for publication in a scientific journal (it would be a perfectly acceptable project/technical report). This is a project summary (line 11-12 on page 2), and the figures are not original, see references to the original works in each of the reference captions.

There are no hypotheses. They do have several goals for the study, but these goals are not generalized beyond the specific county in Arizona in which the work was conducted.

[Figure]

Therefore, it is left to the readers to determine if the work could be used elsewhere.

The authors conducted modeling, but they merely state that modeling was conducted (section 3.2 - 3.3). They do not discuss model details, equations, parameter used in the models, or even input data to the models (e.g. no details on actual rainfall rates used). Again, as openly stated by the authors, this is a summary of a study that was done, it is not an actual scientific research publication with original research.

The discussion merely restates the methods, but does not offer any analysis that would suggest generalization. They made maps to define non-regulatory risk-zone maps. How do we know that these are actual risk zones? For example, they state "... the flood flow zones are probably fairly well constrained." but there is no evidence to persuade readers that their zones are correct. Moreover, the models do not provide compelling evidence to persuade readers that either Laharz or Flo2D are applicable for debris flow runout. There are no tests of the models that show they are correct in the county of interest.

Consequently, I do not think this manuscript is suitable for publication. If the authors were to: create original text (not just a summary) with testable hypotheses, generalizable results, model details, and a logical justification for their models, I think it may then be suitable for publication.

---

## Referee Comment (RC2) · Anonymous Referee #2 · 24 Apr 2019

In the United States, flood and debris-flow hazard assessments are conducted routinely after major wildfires. Such assessments are used by local emergency management officials to identify areas at risk and develop emergency response plans. Often, however, there is insufficient time between the fire and the first rain storm to fully develop emergency response and evacuation plans. This study describes how more complete planning can be achieved by assessing the potential for debris flows before a fire occurs.

The study uses an established fire model to create a wildfire scenario in Coconino County, Arizona, USA. The authors then use the simulated burn severity and a series

of models to evaluate the potential for flooding and debris flow to design rain storms. The hazard assessment includes estimates of flood and debris-flow inundation, which is an analysis that is generally too time consuming to perform during post-fire hazard assessments. There is growing interest in pre-fire hazard assessments, and this the third pre-fire analysis that I am aware of.

The study is clearly valuable for Coconino County, and the discussion of lessons learned during the assessment may appeal to a broader audience. However, I do not think a summary of a published hazard assessment is appropriate for this journal. The manuscript does not test a method/hypothesis or present a significant new concept. I acknowledge that the addition of runout modeling to pre-fire planning is new and important, but this aspect of the manuscript is not fully developed or tested. I think the manuscript would be much stronger if it were reframed to test the proposed methods for pre-fire assessment and demonstrate how well they work. Some questions that could be addressed are: How well does modeled crown fire activity match observed distributions of soil burn severity? How well do the simulated levels of forest treatment reflect burn severity in real fires with real treatments? How accurate are the flood runout predictions? How transferable are estimates of curve numbers from one area to another? How accurate are the estimates of debris-flow probability, volume, and runout? What are the uncertainties? I think this question is particularly important because the predictions of who/what will be impacted will be scrutinized heavily.

In addition, or alternatively, the paper could dig deeper into the planning and mitigation challenges that pre-fire hazard assessments uncover. The end of the paper mentions there were unexpected challenges that came during implementation of the mitigation measures, but these challenges are not described.

Lastly, I think the paper needs to provide more details on the specific methods used in the study. These details are probably included in the engineering reports that the paper references, but more of this information needs to be included in a journal paper to make it easier for readers to understand the assumptions that go into the modeling.

---

## Referee Comment (RC3) · Anonymous Referee #3 · 2 May 2019

Thank you for the opportunity to review this interesting manuscript. It is an original work on an increasingly important problem in environmental management that attempts to assess the potential flooding and debris flow hazards following a wildfire in mountainous northern Arizona and how these hazards might be mitigated by pre-fire fuels treatments. The authors do a good job in the discussion of noting some of the modelling limitations as well as noting some of the social issues in applying this methodology. Overall, this is a well-organized and well-written manuscript, but there are some problems that need to be addressed.

Major issues in no particular order of priority:

• There are many terms that need to be better defined: o On Page 2, how is a 'full-cost accounting' different than an accounting? o On Page 2, what is a 'reasonable-scenario wildfire' as opposed to any other kind of wildfire? The 2010 Schultz Fire is used as a metric throughout this paper because of proximity and the post-fire data available from various studies. The Schultz Fire is commonly referred to as a 'devastating' wildfire. Is the Schultz Fire a 'reasonable-scenario' wildfire? o On Page 2, what does it mean to define the extent and severity of post-wildfire risks at the 'planning-level'? o First occurring on Page 13, what is a 'non-regulatory' risk zone map and how does it differ from a risk zone map? o On Pages 15 and 16, what does it mean to 'increase the resiliency' of the City of Williams should a fire occur? o On Page 16, what is a 'Table Top Exercise'?

• Specific mitigation measures or fuels treatments desperately need to be defined. There are tangential references to mechanical thinning and controlled burning, but nothing about exactly what has changed as a result of these treatments to reduce post-fire hazards. Presumably, biomass has been reduced and fuel loads have been reduced, but how and to what degree? Has stand density been altered? What about stand structure? Dead and down removed? We get no picture of what the landscape will look like, only that the model input parameters have somehow changed. Also, on Page 4, how will the study results and hazard maps identify potential mitigation measures?

• More needs to be presented about the existing vegetation in both Fort Valley and Bill Williams Mountain. We are told generally about the vegetation types in the Colorado Plateau, but virtually nothing about the specific study areas. Fort Valley has a meadow at the bottom and BWM has watersheds with heavily forested slopes, but no other details are provided. Better information must be out there in order to create the burn severity maps, so please share these details with the reader.

• A much better case needs to be developed to equate Crown Fire Activity with Soil Burn Severity. A crown fire will definitely affect residual canopy percentage and tree

mortality, but there are many examples of the soil surface being unaffected by a crown fire. Perhaps something like biomass consumption and energy release along with the height of the canopy would help here. Or just say that 'this is our proxy and we're sticking to it' so the reader can understand the potential limitations.

 c It would be helpful to know if the model FLO-2D PRO has been verified for this area by citing other studies (not your own) where it has been successfully used.

 c The flood modelling uses the 2/10/100 year rainfall events and the debris flow modelling uses the 1/2/5 year rainfall events. It would be helpful to know the magnitude of these values, the length of record that exists, and the methodology by which these storm classes were derived.

 c It is unclear how the runoff curve numbers were assigned. It seems pretty qualitative ('we know there is going to be a lot of runoff, so we will just pick big numbers'). You might also provide a sentence about what this means for the non-specialists who will likely read this paper.

 c For the debris flow modelling, it is unclear how the design storms and the peak I15 relate (if at all).

 c On Page 11, debris flow hazard rankings are based on probability and potential magnitude. More information is needed here to understand how these factors were combined to arrive at these rankings.

 c Why are debris flow volumes of 104.5 and 105 m3 topographically unrealistic (Page 13 and Figure 10)?

Minor issues in no particular order of priority:

 c In some cases, the text is too detailed: o On Page, does it matter that the Schultz Fire burned most of its area in the first 24 hours? o On Page 3, does it matter that the peak directly above Fort Valley is Agassiz Peak? o On Page 3, does it matter that BWM is a cluster of Pliocene dacite, andesite, and benmoreite (whatever that is) domes?  c

On Page 3, does Cataract Creek really originate on the south side of BWM?

â Ăć On Page 5, a reference is needed for the Scott/Reinhardt crown fire calculation method.

â Ăć On Page 6, Line 1, this should be the topic sentence for the next paragraph.

â Ăć In the text, one of the Fort Valley sub-areas is Treated8200; on Figure 4 it is TreatedNW.

â Ăć Table 3 is missing a hyphen in one of the columns.

â Ăć On Page 14, the text labels (Post-fire Debris Flow; Post-fire Hyperconcentrated Flow) do not match those in Figure 11. Also, the legend in Figure 11 is way too small.

â Ăć On Page 16, Lines 8-9, the structural damage that would be sustained in the City of Williams would result from the flooding and erosion.

â Ăć On Page 18, Line 27, the authors had 'not anticipated some of the difficulties local entities have encountered with some of these measures'. Please explain what this means.

Overall this manuscript needs a bit more work. Hopefully these changes will not prove too onerous.

---

## Author Comment (AC1) · 12 Jul 2019

General Comments: The authors appreciate the time and efforts of the three reviewers of this manuscript. The comments and suggestions made in these reviews have helped us to refocus and reframe the manuscript. This requires a significant re-write on our part, and we are continuing to work on the manuscript. Here, we present our rational for why this is an important and original contribution, and the scientific questions we address in this revised manuscript. We then address each reviewers' individual comments. The post-wildfire debris flows and flooding following 2010 Schultz Fire near Flagstaff, Arizona, significantly impacted forest resources, downstream developed ar-

eas and the local economy, which we have described in our paper. This scenario, unfortunately, is not unique to northern Arizona (e.g. Kean et al., 2019; Cannon and Gartner, 2005), nor to the western United States (e.g. Jordan, 2016; Nyman et al., 2015). More densely vegetated forests, longer fire seasons, drought and other climatic influences are expected to contribute to general trends of more frequent and severe wildfires (Kitzberger et al., 2017; Littell et al., 2016; Liu et al., 2013; Krawchuk et al., 2009). This highlights the need for local and regional entities to consider and plan for wildfires and their post-fire impacts to reduce risks and increase community resiliency (Schoennagel et al., 2017).

We hypothesize that risks from post-wildfire debris flows and floods can be assessed, prior to the start of a wildfire, as a function of probability of occurrence, predicted magnitude of flow, and the projected distribution of inundation, and that these data can then be used to identify planning-level risk zones and mitigation opportunities to reduce risks and increase resiliency. Here, we use a post-Schultz Fire dataset that we have compiled over years of working in this area (described below) to test and evaluate the USGS models used to predict the probability of occurrence and magnitude of post-fire debris flows, and Laharz for modeling, prior to a wildfire, potential post-fire debris-flow inundation zones. Most of this work was described in detail in an appendix to the Open File Report (OFR) we referenced in our paper. Here, we describe in more detail that work, and we include a more robust assessment of Laharz by comparing model results with mapped deposits using receiver operator characteristics (ROC) analyses (Fawcett, 2006). We also use our dataset to compare mapped flood inundation areas with modelled FLO-2D inundation zones, again using ROC. Finally, we evaluate the methodology used in this study to assess potential post-fire hazards before a wildfire begins to assess 1) what could be done better, and 2) how other communities could adapt this methodology for their own use.

While the Schultz Fire is only one small fire, the authors, through our continued work on the post-Schultz Fire flows, have a unique dataset of detailed rainfall data, geomorphic

responses of burned basins to rainfall, geomorphic mapping of flood and debris-flow deposits on the piedmont below the burned basins, and 1- and 2-dimensional modeling of design-storm flood flows immediately after the fire and in the years following the fire that are used to inform mitigation efforts and to document post-fire hydrologic recovery. Moreover, there is high resolution (i.e. 1 m) elevation data derived from airborne lidar for our entire study area. Additionally, Coconino County Flood Control District has mapped extents of flood inundation within the burned area and through the downstream developed areas from the July and August, 2010, storms. Therefore, the Shultz Fire presents a rare opportunity to develop and test a methodology that can be more generally applied to assess risks from post-wildfire debris flows and floods.

Reviewer #1: This manuscript does not conform to the stated format of the journal, which asks for "...original research on natural hazards and their consequences." While the writing and figure production is of high quality, this is not an original piece of research and unfortunately is not fit for publication in a scientific journal (it would be a perfectly acceptable project/technical report). This is a project summary (line 11-12 on page 2), and the figures are not original, see references to the original works in each of the reference captions. There are no hypotheses. They do have several goals for the study, but these goals are not generalized beyond the specific county in Arizona in which the work was conducted. Therefore, it is left to the readers to determine if the work could be used elsewhere. The authors conducted modeling, but they merely state that modeling was conducted (section 3.2 - 3.3). They do not discuss model details, equations, parameter used in the models, or even input data to the models (e.g. no details on actual rainfall rates used). Again, as openly stated by the authors, this is a summary of a study that was done, it is not an actual scientific research publication with original research. The discussion merely restates the methods, but does not offer any analysis that would suggest generalization. They made maps to define non-regulatory risk-zone maps. How do we know that these are actual risk zones? For example, they state "... the flood flow zones are probably fairly well constrained." but there is no evidence to persuade readers that their zones are correct. Moreover, the models do

not provide compelling evidence to persuade readers that either Laharz or Flo2D are applicable for debris flow runout. There are no tests of the models that show they are correct in the county of interest. Consequently, I do not think this manuscript is suitable for publication. If the authors were to: create original text (not just a summary) with testable hypotheses, generalizable results, model details, and a logical justification for their models, I think it may then be suitable for publication.

Reply to Reviewer #1: Thank you for your comments. In our revised manuscript we reframe the paper as discussed above. We also address your comments as follows:

1. We are reframing (in progress) the manuscript around the hypothesis that risks from post-wildfire debris flows and floods can be assessed, prior to the start of a wildfire, as a function of probability of occurrence, predicted magnitude of flow, and the projected distribution of inundation, and that these data can then be used to identify planning-level risk zones and mitigation opportunities to reduce risks and increase resiliency.

2. The revised manuscript contains a more comprehensive description of the proposed models for determining flow magnitude and inundation.

3. In order to test our hypothesis, we synthesize the data collected after the Schultz Fire and demonstrate how it was used to test and evaluate the models.

4. Use ROC analyses (Fawcett, 2006) to assess model performances.

5. Provide a discussion of what worked well with the methodology and how we intend to use our methods for two upcoming assessments.

Please also note the supplement to this comment:
https://www.nat-hazards-earth-syst-sci-discuss.net/nhess-2019-74/nhess-2019-74-AC1-supplement.pdf

**Supplement:**

**General Comments:**

The authors appreciate the time and efforts of the three reviewers of this manuscript. The comments and suggestions made in these reviews have helped us to refocus and reframe the manuscript. This requires a significant re-write on our part, and we are continuing to work on the manuscript. Here, we present our rational for why this is an important and original contribution, and the scientific questions we address in this revised manuscript. We then address each reviewers' individual comments.

The post-wildfire debris flows and flooding following 2010 Schultz Fire near Flagstaff, Arizona, significantly impacted forest resources, downstream developed areas and the local economy, which we have described in our paper. This scenario, unfortunately, is not unique to northern Arizona (e.g. Kean et al., 2019; Cannon and Gartner, 2005), nor to the western United States (e.g. Jordan, 2016; Nyman et al., 2015). More densely vegetated forests, longer fire seasons, drought and other climatic influences are expected to contribute to general trends of more frequent and severe wildfires (Kitzberger et al., 2017; Littell et al., 2016; Liu et al., 2013; Krawchuk et al., 2009). This highlights the need for local and regional entities to consider and plan for wildfires and their post-fire impacts to reduce risks and increase community resiliency (Schoennagel et al., 2017).

We hypothesize that risks from post-wildfire debris flows and floods can be assessed, prior to the start of a wildfire, as a function of probability of occurrence, predicted magnitude of flow, and the projected distribution of inundation, and that these data can then be used to identify planning-level risk zones and mitigation opportunities to reduce risks and increase resiliency. Here, we use a post-Schultz Fire dataset that we have compiled over years of working in this area (described below) to test and evaluate the USGS models used to predict the probability of occurrence and magnitude of post-fire debris flows, and Laharz for modeling, prior to a wildfire, potential post-fire debris-flow inundation zones. Most of this work was described in detail in an appendix to the Open File Report (OFR) we referenced in our paper. Here, we describe in more detail that work, and we include a more robust assessment of Laharz by comparing model results with mapped deposits using receiver operator characteristics (ROC) analyses (Fawcett, 2006). We also use our dataset to compare mapped flood inundation areas with modelled FLO-2D inundation zones, again using ROC. Finally, we evaluate the methodology used in this study to assess potential post-fire hazards before a wildfire begins to assess 1) what could be done better, and 2) how other communities could adapt this methodology for their own use.

While the Schultz Fire is only one small fire, the authors, through our continued work on the post-Schultz Fire flows, have a unique dataset of detailed rainfall data, geomorphic responses of burned basins to rainfall, geomorphic mapping of flood and debris-flow deposits on the piedmont below the burned basins, and 1- and 2-dimensional modeling of design-storm flood flows immediately after the fire and in the years following the fire that are used to inform mitigation efforts and to document post-fire hydrologic recovery. Moreover, there is high resolution (i.e. 1 m) elevation data derived from airborne lidar for our entire study area. Additionally, Coconino County Flood Control District has mapped extents of flood inundation within the burned area and through the downstream developed areas from the July and August, 2010, storms. Therefore, the Shultz Fire presents a rare opportunity to develop and test a methodology that can be more generally applied to assess risks from post-wildfire debris flows and floods.

**Reply to Reviewer #1:**

*This manuscript does not conform to the stated format of the journal, which asks for "...original research on natural hazards and their consequences." While the writing and figure production is of high quality, this is not an original piece of research and unfortunately is not fit for publication in a scientific journal (it would be a perfectly acceptable project/technical report). This is a project summary (line 11-12 on page 2), and the figures are not original, see references to the original works in each of the reference captions.*

*There are no hypotheses. They do have several goals for the study, but these goals are not generalized beyond the specific county in Arizona in which the work was conducted. Therefore, it is left to the readers to determine if the work could be used elsewhere.*

*The authors conducted modeling, but they merely state that modeling was conducted (section 3.2 - 3.3). They do not discuss model details, equations, parameter used in the models, or even input data to the models (e.g. no details on actual rainfall rates used). Again, as openly stated by the authors, this is a summary of a study that was done, it is not an actual scientific research publication with original research.*

*The discussion merely restates the methods, but does not offer any analysis that would suggest generalization. They made maps to define non-regulatory risk-zone maps. How do we know that these are actual risk zones? For example, they state "... the flood flow zones are probably fairly well constrained." but there is no evidence to persuade readers that their zones are correct. Moreover, the models do not provide compelling evidence to persuade readers that either Laharz or Flo2D are applicable for debris flow runout. There are no tests of the models that show they are correct in the county of interest.*

*Consequently, I do not think this manuscript is suitable for publication. If the authors were to: create original text (not just a summary) with testable hypotheses, generalizable results, model details, and a logical justification for their models, I think it may then be suitable for publication.*

Thank you for your comments.

In our revised manuscript we reframe the paper as discussed above. We also address your comments as follows:

1. We reframe the manuscript around the hypothesis that risks from post-wildfire debris flows and floods can be assessed, prior to the start of a wildfire, as a function of probability of occurrence, predicted magnitude of flow, and the projected distribution of inundation, and that these data can then be used to identify planning-level risk zones and mitigation opportunities to reduce risks and increase resiliency
2. The revised manuscript contains a more comprehensive description of the proposed models for determining flow magnitude and inundation
3. In order to test our hypothesis, we synthesize the data collected after the Schultz Fire and demonstrate how it was used to test and evaluate the models,
4. Use ROC analyses (Fawcett, 2006) to assess model performances, and
5. Provide a discussion of what worked well with the methodology and how we intend to use our methods for two upcoming assessments.

**Reply to Reviewer #2:**

*In the United States, flood and debris-flow hazard assessments are conducted routinely after major wildfires. Such assessments are used by local emergency management officials to identify areas at risk and develop emergency response plans. Often, however, there is insufficient time between the fire and the first rain storm to fully develop emergency response and evacuation plans. This study describes how more complete planning can be achieved by assessing the potential for debris flows before a fire occurs.*

*The study uses an established fire model to create a wildfire scenario in Coconino County, Arizona, USA. The authors then use the simulated burn severity and a series of models to evaluate the potential for flooding and debris flow to design rain storms.*

*The hazard assessment includes estimates of flood and debris-flow inundation, which is an analysis that is generally too time consuming to perform during post-fire hazard assessments. There is growing interest in pre-fire hazard assessments, and this the third pre-fire analysis that I am aware of.*

*The study is clearly valuable for Coconino County, and the discussion of lessons learned during the assessment may appeal to a broader audience. However, I do not think a summary of a published hazard assessment is appropriate for this journal. The manuscript does not test a method/hypothesis or present a significant new concept. I acknowledge that the addition of runout modeling to pre-fire planning is new and important, but this aspect of the manuscript is not fully developed or tested. I think the manuscript would be much stronger if it were reframed to test the proposed methods for pre-fire assessment and demonstrate how well they work. Some questions that could be addressed are: How well does modeled crown fire activity match observed distributions of soil burn severity? How well do the simulated levels of forest treatment reflect burn severity in real fires with real treatments? How accurate are the flood runout predictions? How transferable are estimates of curve numbers from one area to another? How accurate are the estimates of debris-flow probability, volume, and runout? What are the uncertainties? I think this question is particularly important because the predictions of who/what will be impacted will be scrutinized heavily.*

*In addition, or alternatively, the paper could dig deeper into the planning and mitigation challenges that pre-fire hazard assessments uncover. The end of the paper mentions there were unexpected challenges that came during implementation of the mitigation measures, but these challenges are not described.*

*Lastly, I think the paper needs to provide more details on the specific methods used in the study. These details are probably included in the engineering reports that the paper references, but more of this information needs to be included in a journal paper to make it easier for readers to understand the assumptions that go into the modeling.*

Thank you for your very helpful and constructive comments. We have used your comments as a guide for restructuring the paper. In addition to the above discussion, we address your comments as follows:

1. Your questions regarding how well fire modeling simulates soil burn severity are important and necessary questions. This is an area that sorely needs research. It was, however, beyond the scope of our project, thus we followed previously used procedures to develop our soil burn severity maps. In our upcoming assessments, we plan to use the methods of Staley et al. (2018) to generate historically based burn severity metrics for a high severity fire and a low severity fire that will then be used in our methodology. This is discussed in the discussion section.

2. To better quantify our assessments of model performances, we use ROC analyses to assess mapped deposits and modeled inundation zones. This also helps to define uncertainty levels.
3. The use of curve numbers for assessing the hydrologic impacts of a wildfire is another area of research that is much needed but beyond the scope of this project. For this project, we worked with Coconino National Forest watershed staff and used guidance from the National Resources Conservation Service (2016) to select appropriate curve numbers. This was described in one of the OFR appendices; the revised manuscript has a more detailed description of the choice of this parameter.
4. In the discussion section we elaborate on the unexpected challenges and lessons learned during this project, and our revised steps to help avoid these pitfalls that we will use in two upcoming assessments.
5. We expand our manuscript to provide more detailed descriptions of the models and our methodology.

Again, thank you for the helpful comments.

**Reply to Reviewer #3:**

*Thank you for the opportunity to review this interesting manuscript. It is an original work on an increasingly important problem in environmental management that attempts to assess the potential flooding and debris flow hazards following a wildfire in mountainous northern Arizona and how these hazards might be mitigated by pre-fire fuels treatments. The authors do a good job in the discussion of noting some of the modelling limitations as well as noting some of the social issues in applying this methodology. Overall, this is a well-organized and well-written manuscript, but there are some problems that need to be addressed.*
*Major issues in no particular order of priority:*

Thank you for your helpful and constructive comments. In addition to what we've already outlined above, we are addressing your comments as follows:

- *There are many terms that need to be better defined: o On Page 2, how is a 'full-cost accounting' different than an accounting? o On Page 2, what is a 'reasonable scenario wildfire' as opposed to any other kind of wildfire? The 2010 Schultz Fire is used as a metric throughout this paper because of proximity and the post-fire data available from various studies. The Schultz Fire is commonly referred to as a 'devastating' wildfire. Is the Schultz Fire a 'reasonable-scenario' wildfire? o On Page 2, what does it mean to define the extent and severity of post-wildfire risks at the 'planning level'? o First occurring on Page 13, what is a 'non-regulatory' risk zone map and how does it differ from a risk zone map? o On Pages 15 and 16, what does it mean to 'increase the resiliency' of the City of Williams should a fire occur? o On Page 16, what is a 'Table Top Exercise'?*

1. Thank you for pointing out the use of jargon, confusing and undefined terms. Where possible, we will change the text to plain language. Otherwise we better define our use of unfamiliar terms.
2. Reasonable scenario wildfire – as used here, a likely, high-severity wildfire that can occur under current vegetation and climatic conditions. We will change the text to explain what

that fire would look like (e.g. size, severity) on a given landscape with current fuels and climatic conditions.

3. We will better define non-regulatory risk zone maps which are used to inform and for planning purposes but do not carry any regulatory requirements.

- *Specific mitigation measures or fuels treatments desperately need to be defined. There are tangential references to mechanical thinning and controlled burning, but nothing about exactly what has changed as a result of these treatments to reduce post-fire hazards. Presumably, biomass has been reduced and fuel loads have been reduced, but how and to what degree? Has stand density been altered? What about stand structure? Dead and down removed? We get no picture of what the landscape will look like, only that the model input parameters have somehow changed. Also, on Page 4, how will the study results and hazard maps identify potential mitigation measures?*

- *More needs to be presented about the existing vegetation in both Fort Valley and Bill Williams Mountain. We are told generally about the vegetation types in the Colorado Plateau, but virtually nothing about the specific study areas. Fort Valley has a meadow at the bottom and BWM has watersheds with heavily forested slopes, but no other details are provided. Better information must be out there in order to create the burn severity maps, so please share these details with the reader.*

1. We will include text that better describes a) current conditions above Fort Valley and on Bill Williams Mountain, b) fuels treatments (mechanical biomass reduction, reduced stand densities, controlled burning), and c) the treatments planned for implementation on Bill Williams Mountain.
2. We will better describe how the results from the study were used to identify specific infrastructures at risk from post-fire flows and, where available, the proposed mitigation measures to reduce those risks.

- *A much better case needs to be developed to equate Crown Fire Activity with Soil Burn Severity. A crown fire will definitely affect residual canopy percentage and tree mortality, but there are many examples of the soil surface being unaffected by a crown fire. Perhaps something like biomass consumption and energy release along with the height of the canopy would help here. Or just say that 'this is our proxy and we're sticking to it' so the reader can understand the potential limitations.*

1. Yes, as stated above, this is an area that requires research but was beyond the scope of this project. In our upcoming assessments, we plan to use the methods of Staley et al (2018) to generate historically based burn severity metrics for a high severity fire and a low severity fire that will then be used in our models. This is part of the discussion section.

- *It would be helpful to know if the model FLO-2D PRO has been verified for this area by citing other studies (not your own) where it has been successfully used.*

1. Noted. We will expand the text to include a discussion of other studies and include citations (e.g. Stevens et al., 2011).

- *The flood modelling uses the 2/10/100 year rainfall events and the debris flow modelling uses the 1/2/5 year rainfall events. It would be helpful to know the magnitude of these values, the length of record that exists, and the methodology by which these storm classes were derived.*

1. Noted. The text will be revised to reflect this information which was derived from NOAA Atlas 14.

- *It is unclear how the runoff curve numbers were assigned. It seems pretty qualitative ('we know there is going to be a lot of runoff, so we will just pick big numbers'). You might also provide a sentence about what this means for the non-specialists who will likely read this paper.*

1. The curve numbers were selected with guidance from the Coconino National Forest watershed staff and from the National Resources Conservation Service (2016). This is an area that needs more research but was beyond the scope of this project.

- *For the debris flow modelling, it is unclear how the design storms and the peak I15 relate (if at all).*

1. The text will be revised to explain how the peak I15 was derived for each design storm. Basically, the NOAA Atlas 14 was used to determine I15 for each design storm in the study watersheds.

- *On Page 11, debris flow hazard rankings are based on probability and potential magnitude. More information is needed here to understand how these factors were combined to arrive at these rankings.*

1. This follows the procedures that the USGS uses in assessing post-fire probability and magnitude and was described in detail in one of the OFR appendices. The text will be revised to better explain these models and this method. Appropriate citations will be added.

- *Why are debris flow volumes of 104.5 and 105 m3 topographically unrealistic (Page 13 and Figure 10)?*

1. The volumes of post-fire debris flows are most often derived from channel scour as these debris flows are triggered by runoff, not landslides. The largest post-fire debris flows that have been documented in Arizona are ~ 10,000 $m^3$. The text will be revised to explain this, and citations added.

*Minor issues in no particular order of priority:*
- *In some cases, the text is too detailed: o On Page, does it matter that the Schultz Fire burned most of its area in the first 24 hours? o On Page 3, does it matter that the*

*peak directly above Fort Valley is Agassiz Peak? o On Page 3, does it matter that BWM is a cluster of Pliocene dacite, andesite, and benmoreite (whatever that is) domes?*

1. We will strive to include important details and delete unnecessary information.

- *On Page 3, does Cataract Creek really originate on the south side of BWM?*
  - No, on the north side. Thank you for catching that.

- *On Page 5, a reference is needed for the Scott/Reinhardt crown fire calculation method.*
  1. Done.

- *On Page 6, Line 1, this should be the topic sentence for the next paragraph.*
  1. Done

- *In the text, one of the Fort Valley sub-areas is Treated8200; on Figure 4 it is TreatedNW.*
  1. *Fixed*

- *Table 3 is missing a hyphen in one of the columns.*
  1. Fixed

- *On Page 14, the text labels (Post-fire Debris Flow; Post-fire Hyperconcentrated Flow) do not match those in Figure 11. Also, the legend in Figure 11 is way too small.*
  1. Fixed

- *On Page 16, Lines 8-9, the structural damage that would be sustained in the City of Williams would result from the flooding and erosion.*
  1. Noted.

- *On Page 18, Line 27, the authors had 'not anticipated some of the difficulties local entities have encountered with some of these measures'. Please explain what this means.*
  1. We will expand our text to better describe this.

*Overall this manuscript needs a bit more work. Hopefully these changes will not prove too onerous.*

Thank you again for your helpful comments.

**References**

Cannon, S. H., and Gartner, J. E.: Wildfire-related debris flow from a hazards perspective, in: Debris-flow hazards and related phenomena, edited by: Jakob, M., and Hungr, O., Springer, Berlin, 363-385, 2005.

Fawcett, T.: An introduction to ROC analysis, Pattern Recognition Letters, 27, 861-874, http://dx.doi.org/10.1016/j.patrec.2005.10.010, 2006.

Jordan, P.: Post-wildfire debris flows in southern British Columbia, Canada, International Journal of Wildland Fire, 25, 322-336, http://dx.doi.org/10.1071/WF14070, 2016.

Kean, J. W., Staley, D. M., Lancaster, J. T., Rengers, F. K., Swanson, B. J., Coe, J. A., Hernandez, J. L., Sigman, A. J., Allstadt, K. E., and Lindsay, D. N.: Inundation, flow dynamics, and damage in the 9 January 2018 Montecito debris-flow event, California, USA: Opportunities and challenges for post-wildfire risk assessment, Geosphere, 10.1130/GES02048.1, 2019.

Kitzberger, T., Falk, D. A., Westerling, A. L., and Swetnam, T. W.: Direct and indirect climate controls predict heterogeneous early-mid 21st century wildfire burned area across western and boreal North America, PLoS One, 12, 24, 10.1371/journal.pone.0188486, 2017.

Krawchuk, M. A., Moritz, M. A., Parisien, M.-A., Van Dorn, J., and Hayhoe, K.: Global Pyrogeography: the Current and Future Distribution of Wildfire, PLoS One, 4, e5102, 10.1371/journal.pone.0005102, 2009.

Littell, J. S., Peterson, D. L., Riley, K. L., Liu, Y. Q., and Luce, C. H.: A review of the relationships between drought and forest fire in the United States, Global Change Biology, 22, 2353-2369, 10.1111/gcb.13275, 2016.

Liu, Y., L. Goodrick, S., and A. Stanturf, J.: Future U.S. wildfire potential trends projected using a dynamically downscaled climate change scenario, Forest Ecology and Management, 294, 120-135, http://dx.doi.org/10.1016/j.foreco.2012.06.049, 2013.

Nyman, P., Smith, H. G., Sherwin, C. B., Langhans, C., Lane, P. N. J., and Sheridan, G. J.: Predicting sediment delivery from debris flows after wildfire, Geomorphology, 250, 173-186, http://dx.doi.org/10.1016/j.geomorph.2015.08.023, 2015.

Schoennagel, T., Balch, J. K., Brenkert-Smith, H., Dennison, P. E., Harvey, B. J., Krawchuk, M. A., Mietkiewicz, N., Morgan, P., Moritz, M. A., Rasker, R., Turner, M. G., and Whitlock, C.: Adapt to more wildfire in western North American forests as climate changes, Proceedings of the National Academy of Sciences, 114, 4582-4590, 10.1073/pnas.1617464114, 2017.

Staley, D. M., Tillery, A. C., Kean, J. W., McGuire, L. A., Pauling, H. E., Rengers, F. K., and Smith, J. B.: Estimating post-fire debris-flow hazards prior to wildfire using a statistical analysis of historical distributions of fire severity from remote sensing data, International Journal of Wildland Fire, 27, 595-608, https://doi.org/10.1071/WF17122, 2018.